Accepted at the ICLR 2024 Workshop on AI4Differential Equations In Science

# NEURAL CONTEXT FLOWS FOR LEARNING GENERALIZABLE DYNAMICAL SYSTEMS

**Roussel Desmond Nzoyem**
School of Computer Science
University of Bristol
Bristol, BS8 1QU, UK
rd.nzoyemngueguin@bristol.ac.uk

**David A.W. Barton**
School of Engineering Mathematics and Technology
University of Bristol
Bristol, BS8 1QU, UK
David.Barton@bristol.ac.uk

**Tom Deakin**
School of Computer Science
University of Bristol
Bristol, BS8 1QU, UK
tom.deakin@bristol.ac.uk

## ABSTRACT

Neural Ordinary Differential Equations typically struggle to generalize to new dynamical behaviors created by parameter changes in the underlying system, even when the dynamics are close to previously seen behaviors. The issue gets worse when the changing parameters are unobserved, i.e., their value or influence is not directly measurable when collecting data. We introduce Neural Context Flow (NCF), a framework that encodes said unobserved parameters in a latent context vector as input to a vector field. NCFs leverage differentiability of the vector field with respect to the parameters, along with first-order Taylor expansion to allow any context vector to influence trajectories from other parameters. We validate our method and compare it to established Multi-Task and Meta-Learning alternatives, showing competitive performance in mean squared error for in-domain and out-of-distribution evaluation on the Lotka-Volterra, Glycolytic Oscillator, and Gray-Scott problems. This study holds practical implications for foundational models in science and related areas that benefit from conditional neural ODEs. Our code is openly available at https://github.com/ddrous/ncflow.

## 1 INTRODUCTION

A prototypical dynamical system describes the continuous change of a quantity $x \in \mathbb{R}^d$ through time $t \in \mathbb{R}^+$. Its dynamics are heavily dependent on its parameters $p \in \mathbb{R}^{d_p}$, according to the (ordinary) differential equation[1]

$$\frac{\mathrm{d}x}{\mathrm{d}t}(t) = f(t, x(t), p), \tag{1}$$

where $f : \mathbb{R}^+ \times \mathbb{R}^d \times \mathbb{R}^{d_p} \to \mathbb{R}^d$ is the **vector field**. Learning a dynamical system from data is synonymous with approximating $f$, a task neural networks have been remarkably good at in recent years (Chen et al., 2018; Rackauckas et al., 2020; Kidger, 2022).

Neural Ordinary Differential Equations (neural ODEs) (Chen et al., 2018) have emerged as a versatile class of methods for learning ordinary, stochastic, and controlled differential equations (Kidger et al., 2020). They can be viewed as continuous-time limits of Residual Neural Networks (He et al., 2016; Haber & Ruthotto, 2017; Weinan, 2017), and are sometimes referred to as Universal Differential Equations (Rackauckas et al., 2020). Trained using Differentiable Programming techniques (Nzoyem Ngueguin et al., 2023; Ma et al., 2021), they have shown extensive applications in areas like chemical engineering (Owoyele & Pal, 2022), geosciences (Shen et al., 2023), turbulence forecasting (Portwood et al., 2019), etc., where the parameters are typically known or the environments fixed.

---

[1]The dynamical system determined by one parameter value $p$ will henceforth be called an "environment".

In Scientific Machine Learning (Cuomo et al., 2022), the problem of generalization has largely been tackled by injecting domain knowledge. It is commonly understood that adding a term in eq. (1) that captures as much of the dynamics of the problem as possible leads to lower evaluation losses (Yin et al., 2021b). For such terms to be added, however, it is essential to have knowledge of the parameters that change, which may then either be directly estimated, or predicted by a neural network within the vector field during training (Rackauckas et al., 2020). We are naturally left to wonder how to efficiently learn a generalizable dynamical system when such physics are absent.

The two major problems to learning the parameter dependence of a vector field are:

(P1) **Limited data**: models like neural ODEs are known to be data-hungry (Yin et al., 2021b), and limited data from each environment may not be enough to learn a vector field suitable for all environments.

(P2) **Unobserved parameters**: during the data collection process, one might be unfamiliar with the basic physics of the system, or may not know *which* or *how many* parameters might be worth paying attention to.

Solving these two problems would contribute to the efficient **generalization** of the learned models; a task attempted, in the context of dynamical systems, by several methods in recent years. LEADS (Yin et al., 2021a) is a Multi-Task Learning method that decomposes the vector field into shared and environment-specific components. In the realm of Meta-learning, DyAd (Wang et al., 2022) uses an encoder network to predict time-invariant latent context vectors (or simply **contexts**), then a forecaster to generate trajectories. Similar to DyAd, FOCA (Park et al., 2023) leverages Meta-learning in a bi-level optimization process, the cost of which is lowered by making use of exponential moving averages (EMAs) to avoid second-order derivatives. The state-of-the-art Meta-learning method is CoDA (Kirchmeyer et al., 2022) which applies the contexts directly to the neural network weights. Its main limitation is the use of a hypernetwork, which may hinder parallelism and lead to a complex optimization landscape (Chauhan et al., 2023). Further details concerning these methods and how they relate to ours can be found in appendix A.

In this paper, we propose Neural Context Flows, a new method that uses continuity and differentiability of the vector field wrt the unobserved parameters as an inductive bias. All the environments share the same neural network weights, but are encoded with different latent context vectors to modulate the behavior of the neural network. This is achieved via first-order Taylor expansion of the vector field around the context vector, a linearization that imposes a smooth context landscape and easily reconstructs dynamical systems from multiple environments. The resulting method is easily extendable and parallelizable across environments. Our contribution is this novel framework for generalizable dynamical systems learning, thus providing an answer for (P1) and (P2).

## 2 METHOD

Given sample trajectories $\{x_{0:N}^e\}_{1 \le e \le m}$ of length $N \in \mathbb{N}^*$ collected[2] from $m$ distinct but related environments, our goal is to find the weights $\theta$ and contexts $\{\xi^e\}_{1 \le e \le m}$ that condition a vector field

$$\frac{\mathrm{d}x^e}{\mathrm{d}t}(t) = f_\theta(t, x^e(t), \xi^e), \qquad \forall e \in [\![1, m]\!].\tag{2}$$

We learn a single vector field for all environments in our training set $\mathcal{D}_{\mathrm{tr}}$. The same vector field will be reused, unchanged, for future testing and adaptation to environments in $\mathcal{D}_{\mathrm{te}}$ and $\mathcal{D}_{\mathrm{ad}}$. Similar to equation 1, the vector field $f_\theta$ is assumed to not only be *continuous*, but also *differentiable* wrt its third argument, $\xi$. Exploiting this inductive bias, we replace $f_\theta$ with $T_{f_\theta}$, its first-order Taylor expansion around any other $\{\xi^j\}_{0 \le j \le m}$, giving rise, for a fixed $e$, to $m$ neural ODEs

$$\begin{cases} x^{e,j}(0) = x_0^e, \\ \dfrac{\mathrm{d}x^{e,j}}{\mathrm{d}t}(t) = T_{f_\theta}(t, x^{e,j}(t), \xi^e, \xi^j), \end{cases} \quad \forall j \in [\![1, m]\!] \tag{3}$$

---

[2]We may have multiple trajectories per environment, all generated from an unknown distribution $\mathcal{D}$ which may change depending on whether we are performing training, in-domain testing, or OOD adaptation.

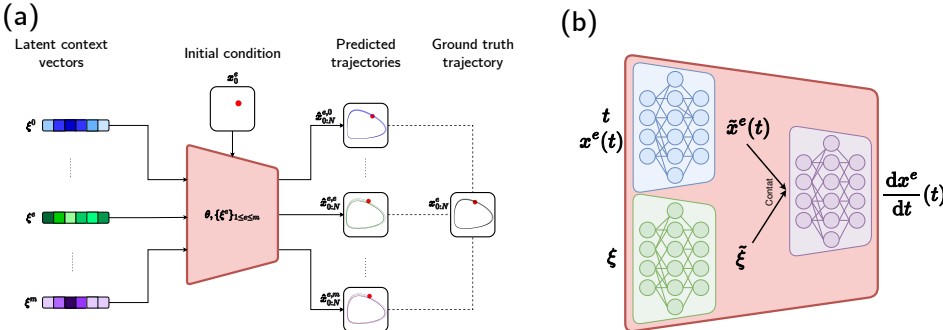

**FIGURE 1: Illustration of the Neural Context Flow (NCF). (a) Given one initial condition for a specific environment $x_0^e$, NCF predicts, in parallel, $m$ candidate trajectories. (b) 3-networks architecture of $f_\theta$.**

where, for $t \in \mathbb{R}^+, x^{e,j} \in \mathbb{R}^d$, and $\xi^e, \xi^j \in \mathbb{R}^{d_\xi}$,

$$
\begin{aligned}
T_{f_\theta}(t, x^{e,j}, \xi^e, \xi^j) = &\, f_\theta(t, x^{e,j}, \xi^j) \\
&+ \langle \nabla_\xi f(t, x^{e,j}, \xi^j), \xi^e - \xi^j \rangle \\
&+ o(\|\xi^e - \xi^j\|).
\end{aligned} \tag{4}
$$

During training (as described below) trajectories from each of these $m$ neural ODEs are used within the loss function built upon

$$
\mathcal{L}(\hat{x}_{0:N}^{e,j}, x_{0:N}^e) = \frac{1}{N \times d} \sum_{n=0}^{N-1} \|\hat{x}_n^{e,j} - x_n^e\|_2^2 + \|\xi^e\|_1, \tag{5}
$$

where $\hat{x}_{0:N}^{e,j}$ is a predicted trajectory from eq. (3) using a differentiable numerical integrator, and $\|\cdot\|_1$ is the Lasso regularization (Tibshirani, 1996) to enforce sparsity in the potentially high-dimensional context vector. We call this new framework **Neural Context Flow** (NCF), referring to the capability of the context from one environment, i.e., $j$, to influence trajectories from another environment, i.e., $e$, during training and adaptation. Fig 1a illustrates this process of generating up to $m$ candidate trajectories, all based on different context vectors, to be compared to a single ground truth trajectory. In fig. 1b, we highlight the **3-networks** architecture that improves robustness of NCFs. The context and state networks (in green and blue respectively), bring the environment-specific context vector $\xi$ and the state $x^e$ into the same representational space before concatenating and processing by the main network (in purple).

**Training** Training is performed using gradient descent, where we alternate between updates of the neural network weights $\theta$, and the contexts $\{\xi^e\}_{1 \le e \le m}$ (see algorithm 1).

---

**Algorithm 1** Training of Neural Context Flows

---

1: **Input:** $\xi_{\text{tr}} = [\![1, m]\!] \subset \xi, \{\mathcal{D}_{\text{tr}}^e\}_{e \in \xi_{\text{tr}}}$ with $\forall e \in \xi_{\text{tr}}, |\mathcal{D}_{\text{tr}}^e| = D_{\text{tr}}$
2: $\theta \in \mathbb{R}^{d_\theta}$ randomly initialized
3: $\xi^{1:m} = \bigcup_{e \in \xi_{\text{tr}}} \xi^e$, where, $\forall e \in \xi_{\text{tr}}, \xi^e = \mathbf{0} \in \mathbb{R}^{d_\xi}$
4: **repeat**
5: $\qquad \mathcal{L}_{\text{tot}}(\theta, \xi^{1:m}) = \dfrac{\sum_{e=1}^m \sum_{x_{0:N}^e} \sum_{j=1}^m \mathcal{L}(\hat{x}_{0:N}^{e,j}, x_{0:N}^e)}{m^2 \times D_{\text{tr}}}$
6: $\qquad \theta \leftarrow \theta - \nabla_\theta \mathcal{L}_{\text{tot}}(\theta, \xi^{1:m})$
7: $\qquad \xi^{1:m} \leftarrow \xi^{1:m} - \nabla_{\xi^{1:m}} \mathcal{L}_{\text{tot}}(\theta, \xi^{1:m})$

---

**Testing** During testing, the same vector field in eq. (3) is considered, but only for $j = e$ (thus returning to a standard neural ODE). That said, the underlying parameters of the dynamical systems are unchanged, and so the learned contexts are reused. As for the data used, only the initial conditions (and hence the resulting trajectories) are unseen. We call this type of evaluation **In-Domain** evaluation.

**Adaptation** Adaptation to new environments requires relatively fewer iterations. We perform the same gradient descent steps as during training, with the only difference that the network weights are now fixed, and only the contexts are optimized. In pseudo-code, the training-specific inputs in algorithm 1 are replaced with those for adaptation ($\xi_{ad}, \mathcal{D}_{ad}$, etc.), line 2 is adapted, and line 6 is removed. During this stage, both the underlying parameters and the initial conditions are unseen. We refer to testing during adaptation as **OOD** evaluation.

**TABLE 1: In-domain (InD) and adaptation (OOD) test MSEs ($\downarrow$) for the Lotka-Volterra (LV) and Gycolytic Oscillator (GO) problems. The best is reported in bold. Ours is shaded in gray.**

|  | **LV** ($\times 10^{-5}$) | | | **GO** ($\times 10^{-3}$) | | |
|  | #Params | InD | OOD ($\times 10^{-5}$) | #Params | InD | OOD |
|---|---|---|---|---|---|---|
| CAVIA | 305714 | 91.0$\pm$63.6 | 120.1$\pm$28.3 | 130303 | 6.40$\pm$1.41 | 46.3$\pm$8.49 |
| CoDA | 305793 | **1.40**$\pm$0.13 | **2.19**$\pm$0.78 | 135390 | **0.56**$\pm$0.08 | **0.42**$\pm$0.43 |
| NCF | 308240 | 9.56$\pm$2.01 | 9.83$\pm$1.10 | 131149 | 4.03$\pm$0.91 | 1.94$\pm$0.12 |

**TABLE 2: In-domain (InD) and adaptation (OOD) test MSEs ($\downarrow$) for the Gray-Scott (GS) problem. The best is reported in bold. Ours is shaded in gray.**

|  | **GS** ($\times 10^{-3}$) | | |
|  | #Params | InD | OOD |
|---|---|---|---|
| CAVIA | 618245 | 69.9$\pm$21.2 | 68.0$\pm$4.2 |
| CoDA | 619169 | **1.23**$\pm$0.14 | **0.75**$\pm$0.65 |
| NCF | 610942 | 7.64$\pm$0.70 | 5.57$\pm$0.21 |

Neural Context Flows are strongly parallelizable and extendable (as evidenced in line 5 of algorithm 1). Besides those properties, NCFs are attractive because of their empirical performance on practical problems, which we highlight in the next section.

## 3 RESULTS

Our goal is to evaluate how our framework compares to alternative Multi-Task and Meta-Learning approaches. To that end, we consider the Lotka-Volterra (LV) problem (Bacaër & Bacaër, 2011) which models the evolution of the concentration of preys and predators in a closed ecosystem. The behavior of this system is controlled by four parameters: $\alpha$, $\beta$, $\gamma$, and $\delta$. We repeat the experiment as designed in (Kirchmeyer et al., 2022). All synthetic ground truth data is generated with the initial states following the Unif$(1, 3)$ distribution. The parameters that vary across training environments are $\beta \in \{0.5, 0.75, 1\}$ and $\delta \in \{0.5, 0.75, 1\}$. In each training environment, we generate 4 trajectories with a Runge-Kutta 4th order scheme, while we generate 32 for in-domain evaluation. For adaptation, we extrapolate to $\beta \in \{0.625, 1.125\}$ and $\delta \in \{0.625, 1.125\}$, with only 1 trajectory per environment for training, and 32 for OOD evaluation. The observed parameters $\alpha$ and $\delta$ are always fixed at $0.5$. We present additional details on the experimental setup and its implementation with NCFs in appendix C.

Additionally, we consider the Glycolytic-Oscillator (GO) ODE, and the Gray-Scott (GS) PDE on a 2D grid with periodic boundary conditions. The GO system models rhythmic fluctuations in chemical activity, while GS models reaction-diffusion processes often observed in pattern formation in biological and chemical systems. Details of the experimental setup can be found in (Kirchmeyer et al., 2022). The LV, GO and GS datasets were recreated in an uniform manner to federate reproducibility efforts in the area of data-driven dynamical systems learning. We have open-sourced the datasets at `https://github.com/ddrous/gen-dynamics`.

Our method is compared to two baseline methods known for their compelling meta-learning capabilities: CAVIA (Zintgraf et al., 2019), and CoDA with $l_1$ context regularization (Kirchmeyer et al., 2022). As a gradient-based meta-learning method, CAVIA is the most similar to ours, but only CoDA is specifically designed for dynamical systems. We disregard DyAD (Wang et al., 2022)

for comparison since it requires an initial window for encoding; whereas here, we only assume knowledge of the initial state.

Results from tables 1 and 2 indicate that CAVIA struggles to meta-learn all dynamical systems and to adapt to new scenarios. At similar neural network sizes, CoDA achieves the best MSEs for in-domain and out-of-distribution evaluations. As for Neural Context Flows, we notice a competitive second-best metric on all problems.

In addition to the dynamics forecasting experiments above, we perform ablation studies to clarify the roles of the context vector and the 3-networks architecture in our framework. We observe that using larger context vectors allows more freedom for NCF to adjust the behavior of the neural networks on each environment of interest (appendix B.2.1). Similarly, removing the context and state networks as described in fig. 1b considerably lowers the performance of our method (appendix B.2.2).

## 4 DISCUSSION

**Limitations**   While endlessly extendable and parallelizable, Neural Context Flows still need to compute several candidate trajectories during training. As such, NCFs incurs a quadratic cost $O(m^2)$ in the number of meta-training environments. With a massive increase in $m$, this poses not only runtime, but also memory restriction on the scaling of NCFs. We address this issue in future work by randomly sampling from a fixed pool $\mathcal{P}$ only a few neighboring environments for Taylor expansion in eq. (4). This preserves accuracy while reducing the computational workload.

Neural Context Flows also lack theoretical backing to explain their surprising effectiveness. For instance, an algorithmic analysis could provide guarantees that simulated trajectories in different environments never collapse into each other as a consequence of the context vectors from those environments converging to the same value. This can be addressed by implementing more robust alternating minimization algorithms such as (Attouch et al., 2010), which we leave for future work.

**Conclusion**   We introduced Neural Context Flows (NCFs), a new framework for training models that easily generalize across environments by leveraging continuity and differentiability of the vector field with respect to its arguments. These are two weak assumptions that are typically satisfied with dynamical systems encountered in physical sciences. Our framework is experimentally validated with three benchmark problems commonly used in the field. NCFs compare well to existing approaches in terms of in-domain and out-of-distribution test metrics. This method is not only a promising step towards generalizing dynamical systems under limited data, but a fresh perspective into the conditioning of machine learning models in general.

ACKNOWLEDGMENTS

This work was supported by UK Research and Innovation grant EP/S022937/1: Interactive Artificial Intelligence, EPSRC program grant EP/R006768/1: Digital twins for improved dynamic design, EPSRC grant EP/X039137/1: The GW4 Isambard Tier-2 service for advanced computer architectures, and EPSRC grant EP/T022205/1: JADE: Joint Academic Data science Endeavour-2.

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

# A    EXTENDED RELATED WORK

Using purely data-driven models to learn ODEs that generalize across parameters is only a recent endeavor (Yin et al., 2021a; Wang et al., 2022; Kirchmeyer et al., 2022; Park et al., 2023). To the best of the author's knowledge, all attempts to solve (P1) and (P2)[3] from section 1 have relied on Multi-Task and Meta-Learning to efficiently adapt to new parameter values, thus producing methods with varying levels of interpolation and extrapolation capabilities.

**Multi-Task Learning**    Multi-task Learning (MTL) describes a family of techniques where a model is trained to jointly perform multiple tasks. In Scientific Machine Learning (Cuomo et al., 2022), one of the earliest methods to attack this generalization problem is called LEADS (Yin et al., 2021a). In LEADS, the vector field is decomposed into shared dynamics $f_\phi$ and environment-specific $g_\psi^e$ components

$$\frac{\mathrm{d}x^e}{\mathrm{d}t} = f_\phi(t, x) + g_\psi^e(t, x), \tag{6}$$

where the superscript $e$ identifies the environment in which the dynamical system evolves, and $\{\phi, \psi\}$ are learnable neural network weights.

While LEADS works well at interpolation tasks, it performs poorly at extrapolation (Kirchmeyer et al., 2022). Furthermore, it requires retraining a new network $g_\theta^e$ each time a new environment is encountered, which can be limiting in scenarios where adaptation is frequently required. Before LEADS, other MTL approaches had been proposed outside the context of dynamical systems (Caruana, 1997; Rebuffi et al., 2017; 2018). Still, none of these address the crucial adaptation to new tasks which is our focus. As such, it is unsurprising that they all encounter challenges during extrapolation.

**Meta-Learning**    Another influential body of work looked at Meta-learning (Hospedales et al., 2021; Finn et al., 2017): a framework in which, in addition to the MTL joint training scheme, shared representation are learned in order to fast adapt to unseen tasks with only minimum data (Wang et al., 2021).

One of the first Meta-learning approaches to look at generalizable dynamical systems is DyAd (Wang et al., 2022). It learns to represent time-invariant features of a trajectory by an *encoder* network, followed by a *forecaster* network to learn shared dynamics across the different environments. In DyAd, two layers (AdaIN and AdaPad) are introduced to allow the encoder to accurately guide the (convolutional) forecaster. That said, DyAd only performs well under weak supervision; that is, when known, the underlying parameters are made known to the loss function by penalizing their residual norm with a linear mapping applied to the time-invariant context. Similar to DyAD, FOCA (Park et al., 2023) leverages Meta-learning in a bi-level optimization process, the cost of which is lowered by making use of exponential moving averages (EMAs) to avoid second-order derivatives.

Arguably the most successful method in Meta-learning is CoDA (Kirchmeyer et al., 2022). Unlike typical gradient-based Meta-learning (GBML) methods (Finn et al., 2017), CoDA assumes that the underlying system is described by a parametrized differential equation whose form is shared by all environments. However, those equations differ by the values of the vector field's weights, which are produced by a *hypernetwork*. The weights for environment $e$ are computed by[4]

$$\theta^e = \theta^c + W\xi^e, \tag{7}$$

where $\theta^c$ and $W$ are shared across environments, $\xi^e \in \mathbb{R}^{d_\xi}$ is an environment-specific latent context vector (or simply **context**). These considerations allow CoDA to achieve state-of-the-art performance on Lotka-Volterra, Glycolytic-Oscillator, Gray-Scott, and Navier-Stokes forecasting problems (Kirchmeyer et al., 2022). The main limitation of CoDA is its hypernetwork approach, which limits parallelism and increases computational cost. In practice, since they involve two networks instead of one, methods based on hypernetworks require more computational resources to train, and exhibit a more complex optimization landscape (Chauhan et al., 2023).

---

[3]We focus our literature review on solving (P1) and (P2), within the context of dynamical systems. For a more exhaustive review, see (Kirchmeyer et al., 2022).

[4]MTL models can be identified to eq. (7) with $\theta^c = \mathbf{0}$ since they remove the ability of performing fast adaptation when parameters are retrained from scratch.

**TABLE 3: In-domain and adaptation evaluation MSE values on the SP dynamics forecasting problem, for One-For-All (OFA) and Neural Context Flow (NCF) techniques. Note the difference in exponents: $0$ for OFA, and $-3$ for NCF.**

|  | IN-DOMAIN | ADAPTATION |
| --- | --- | --- |
| OFA ($\times 10^0$) | $1.14 \pm 0.002$ | $3.31 \pm 0.026$ |
| NCF ($\times 10^{-3}$) | $4.82 \pm 0.08$ | $0.25 \pm 0.02$ |

Our method, by leveraging mathematical insights from Taylor expansion and differential calculus, is more related to Meta-learning, and makes the same underlying assumptions as CoDA (Kirchmeyer et al., 2022). But unlike CoDA, ours is not limited by a hypernetwork; this means that it is not only more accurate, but also more straightforward in terms of implementation and optimization.

## B   ADDITIONAL EXPERIMENTS

Through these additional experiments, we aim to answer two main questions:

1. How well do NCFs solve the generalization problem with limited data in a physics-agnostic setting ? (see appendix B.1)

2. How does its context size and its 3-networks architecture affect the performance of NCFs ? (see appendix B.2)

### B.1   SIMPLE PENDULUM (SP)

In addition to the Lotka-Volterra (LV) systems from section 3, the Neural Context Flow framework is evaluated on the simple undamped pendulum (SP). The SP experiment highlights the benefits of using NCFs in lieu of baseline approaches like One-For-All (OFA), where one context-agnostic vector field is learned for all environments indiscriminately.[5]

The autonomous dynamical system at play here corresponds to a frictionless pendulum swinging from a stationary point. The state space $x = (\alpha, \omega)$, comprises the angle the pendulum makes with the vertical, and its angular velocity, respectively

$$\begin{cases} \dfrac{\mathrm{d}\alpha}{\mathrm{d}t} = \omega, \\ \dfrac{\mathrm{d}\omega}{\mathrm{d}t} = -\dfrac{g}{L}\sin(\alpha). \end{cases} \tag{8}$$

For this problem, each environment corresponds to a different gravity[6], to which we don't have access during the data collection stage. Therefore, with $L = 1$ completely known, the goal is to learn a dynamical system that easily generalizes across the unobserved $g$.

During training, we use 25 environments with $g$ regularly spaced in $[2, 24]$. Each of these environments contains only 4 trajectories with the initial conditions $x^e(0) \sim (\mathrm{Unif}(-\frac{\pi}{3}, \frac{\pi}{3}), \mathrm{Unif}(-1, 1))$. For testing, the same environments are used, and the same initial condition distribution is used. During adaptation, we interpolate to 2 new environments where $g \in \{10.25, 14.75\}$. We expand on this experimental setup in appendix C.

Our mean squared error (MSE) loss values during training is reported in fig. 2. The in-domain and adaptation metrics are reported in table 3 as we carry out a comparison with the baseline One-For-All (OFA). Unsurprisingly, learning one context-agnostic vector field for all environments (OFA) is suboptimal given the differences in gravity from one environment to the next. On the other hand, our method learns to discriminate between environments, and produces accurate trajectories.

---

[5]We refer the reader to (Yin et al., 2021a) for a deeper discussion on One-For-All and One-Per-Env (one vector field for each new environment), and how they relate to the current problem.

[6]To give an intuition behind the term "environment", one might consider the surface of a celestial body in the solar system (see section 1).

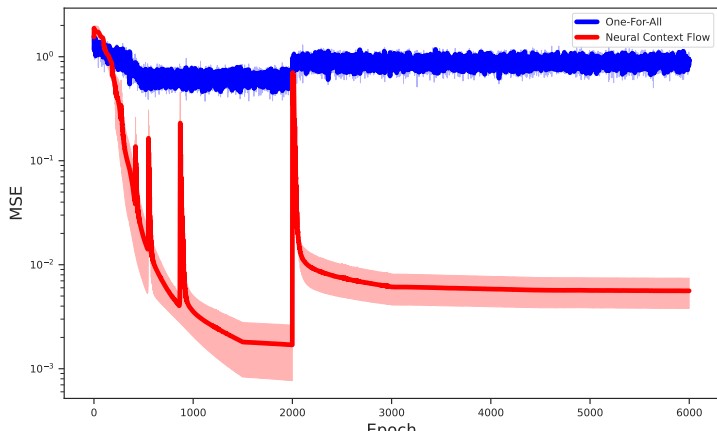

**FIGURE 2: MSE loss values when training a Neural Context Flow on the SP problem, compared with a baseline One-For-All formulation. OFA fails catastrophically, since the diversity of environments in the training dataset prevents the approximation of any meaningful vector field.**

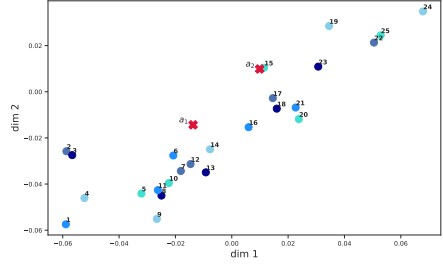

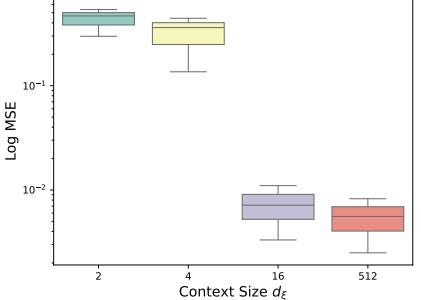

**FIGURE 3: Representation of the first and second dimensions of the learned contexts for the SP problem. The labels 1 to 25 identify the training environments, while $a_1$ and $a_2$ (in red), indicate the adaptation environments.**

**FIGURE 4: In-Domain evaluation for the SP problem as the context size is increased.**

Beyond serving as a control signal for the vector fields, the contexts encode useful representations. In fig. 6, we visualize the first two dimensions of the various $\{\xi^e\}_{1 \le e \le 25}$ after training. We observe that environments close in indices[7] are equally close in the two context dimensions. Similarly, distant environments are noticeably far apart in context space. The same observation is made during adaptation, where, for instance, $e' = a_2$ (corresponding to $g = 14.75$) gets a context close to $e = 15$ (corresponding to $g = 14.83$). This observation indicates that the latent context vector is encoding features related to gravity, which may be used for further downstream representation learning tasks.

---

[7]The indices of the training environments correspond to their ordering in increasing values of gravity.

**TABLE 4: MSE upon ablation of the 3-networks architecture (NCF*) on both SP and LV problems, highlighting difficulties adapting to new environments.**

| | SP ($\times 10^{-3}$) | | LV ($\times 10^{-5}$) | |
|---|---|---|---|---|
| | IN-DOMAIN | ADAPTATION | IN-DOMAIN | ADAPTATION |
| NCF | $4.82 \pm 0.08$ | $0.25 \pm 0.02$ | $1.28 \pm 0.25$ | $0.8 \pm 1.01$ |
| NCF* | $8.39 \pm 0.009$ | $1239.64 \pm 867.6$ | $11.03 \pm 1.1$ | $69.98 \pm 84.36$ |

## B.2 ABLATION STUDIES

### B.2.1 RESTRICTION OF THE CONTEXT SIZE

The latent context vectors are the building blocks of Neural Context Flows. Having showed that they encode useful representations vital for downstream tasks in appendix B.1, we now inquire as to how their size influences the overall learning performance.

Like CoDA (Kirchmeyer et al., 2022), the context size $d_\xi$ is directly related to the parameter count of the model; and limiting the parameter count bears practical importance for computational efficiency and interpretability. Thus, we perform the SP experiment as described in appendix B.1 with $d_\xi \in \{2, 4, 16, 512\}$. The results observed in fig. 3 align with our intuitive understanding of increased expressiveness with bigger latent vectors.

More importantly, fig. 3 indicates that while there is little performance to be gained from setting $d_\xi = 16$ rather than 512, there is a significant loss of information going down from 16 to 4, a value below which the NCF essentially fails to learn any meaningful dynamic. This observation holds particular significance due to the underlying dynamical system only having one variable parameter: its gravity $g$. This finding highlights the complexity of the relationship between $\xi$ and $g$, suggesting that NCFs could potentially benefit from the vast body of research in representation learning.

### B.2.2 ABLATION OF THE 3-NETWORKS ARCHITECTURE

Another key element of the NCF framework is the 3-networks architecture described in fig. 1b. Like the context size, its ablation directly contributes to reduction in parameter count. For this experiment, we remove the data- and context-specific neural networks in the vector field, and we directly concatenate $t, x^e(t)$, and $\xi$; the result of which is passed to a (shared) neural network.

This study is of particular significance since, if in addition to foregoing the 3-networks architecture, we eliminated the Taylor expansion step, and NCF would transform into the well-established data-controlled neural ODE (Massaroli et al., 2020). So we run both dynamics forecasting problems without the 3-networks architecture, and we report the training MSE in fig. 5. In-domain and adaptation MSEs for both LV and SP problems are reported in table 4.

While fig. 5 highlights a performance discrepancy of nearly 1 order of magnitude during training, the key insight is hidden in the adaptation columns of table 4. Indeed, the removal of the data- and context-specific networks in the vector field considerably restricts the model's ability to generalize to unseen environments, both for the SP and LV problems. This consolidates the 3-networks architecture as an essential piece of the NCF framework.

## C EXPERIMENTAL DETAILS

In this section, we share crucial details that went into the experiments conducted in section 3 and appendix B.2.

**Details specific to NCFs** We use the 3-networks architecture from fig. 1b: three separate MLPs to process the state variable, the context, and the concatenation of encodings of both. The context size is set to $d_\xi = 1024$ for SP, LV, and GO problem; and $d_\xi = 256$ for GS. The regularization function from eq. (5) is the $L_1$ over the contexts. We make use of two distinct Adabelief optimizers (Zhuang et al., 2020): one for the neural network weights, another for the contexts.

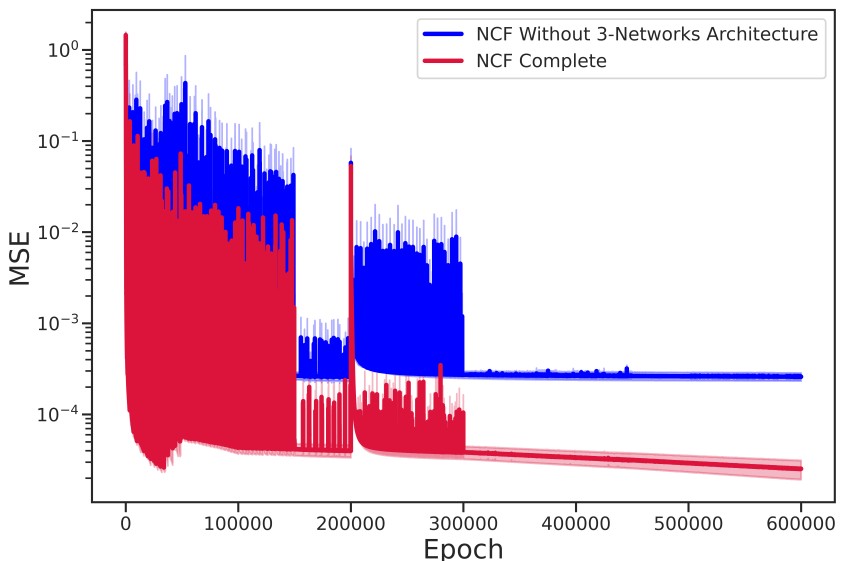

**FIGURE 5: MSE loss curves when training the LV problem on a complete NCF, and on NCF\*, the NCF variant deprived of the 3-networks architecture.**

**Model architecture**   All neural networks employed in SP, LV, and GO are MLPs with `swish` activation functions, and dense layers made up of $[s_\text{in} \to s_0 \to 64 \to s_\text{out}]$ neurons. For the state MLP (in blue in fig. 1b), we had $s_0 = 64$, and $s_\text{out} = 64$. We used $s_\text{in} = 2$ for SP and LV, $s_\text{in} = 7$ for GO. Due to the large context size employed, the context network (in green in fig. 1b) had $s_\text{in} = 1024$, $s_0 = 264$, and $s_\text{out} = 64$. This allowed both the data and the contexts to be encoded into similar representations, before concatenation and passing to the shared neural network (in purple in fig. 1b), which had $s_\text{in} = 128$, $s_0 = 64$, and $s_\text{out} = 2$ for SP and LV, or $s_\text{out} = 7$ for GO. Concerning the Gray-Scott PDE, we used a 1-layer context CNN composed of a dense layer of $[256 \to 2048]$ followed by a convolutional layer with 8 channels. We also used a 1-layer state CNN with 8 channels, and a 4-layers shared CNN with 64 channels. All kernels were of size $3 \times 3$.

**Training scheme**   Training a neural ODE can be subject to considerable pitfalls, the first one being *stability*. Despite the danger of falling into local minima (Kidger et al., 2020), we directly trained our models on 100% of the trajectories. This straightforward strategy was possible because our trajectories typically had small lengths (10 to 20 time steps). As for the propagation of gradients from the loss function to the learnable parameters in the right-hand-side of the neural ODEs (3), we decided to differentiate through our solvers, rather than numerically computing adjoints (Chen et al., 2018). Since our main solvers were the less demanding RK4 and Dopri5 solvers, we didn't incur the heavy memory cost Differentiable Programming methods are typically known for (Nzoyem Ngueguin et al., 2023; Kidger et al., 2020).

**Training hyperparameters**   We used the full-batch gradient descent approach for all experiments, which sped up the training processes. With the aforementioned training scheme, the total number of training epochs was 6k for SP, 120k for LV, 2.4k for GO, and 10k for GS. The number of adaptation epochs was 2k for SP, and 1.5k for LV, GO, and GS. As for the learning rate schedule, we typically started at $10^{-4}$, then divided by 10 after 25% of the total training epochs had been completed; again by 10 after 50%; and one final time at 75% completion.

**Data generation**   For all experiments, the data is synthetic, with the parameters and initial states sampled from distributions representative of problems faced in the scientific community. This is a common practice in this emerging field of generalizable dynamical systems, where the search for unified benchmarks and datasets is still an open-problem (Massaroli et al., 2020; Kirchmeyer et al., 2022).

## D    EXAMPLE IMPLEMENTATION OF NCFS

**Vector field**    The vector field takes center-stage when modelling using ODEs. For a highly performant implementation of neural ODEs, we leveraged JAX (Bradbury et al., 2018) and its ever-growing ecosystem; in particular Optax (DeepMind et al., 2020) for optimization, and Equinox (Kidger & Garcia, 2021) for neural network definition. The Jacobian-vector product primitive `filter_jvp` as illustrated below, coupled with `jit`-compilation, allowed commendable runtimes for both problems.

```
1  class ContextFlowVectorField(eqx.Module):
2      physics: eqx.Module
3      augmentation: eqx.Module
4
5      def __init__(self, augmentation, physics=None):
6          self.augmentation = augmentation
7          self.physics = physics if physics is not None else NoPhysics()
8
9      def __call__(self, t, x, ctx, ctx_):
10         # ctx = \xi^e, and ctx_ = \xi^j
11
12         vf = lambda xi_: self.physics(t, x, xi_) \
13                         + self.augmentation(t, x, xi_)
14         gradvf = lambda xi_, xi: eqx.filter_jvp(vf, (xi_,), (xi-xi_,))[1]
15
16         return vf(ctx_) + gradvf(ctx_, ctx)
```

LISTING 1: **Taylor expansion of the NCF vector field**

**Loss function**    The loss functions in Neural Context Flows is defined in two stages. The outer loss function roughly corresponds to line 5 in algorithm 1, not including the innermost summation across the indices $j$, which is taken care of by the inner loss. The vectorized implementation of the inner loss function is defined below. It highlights our $L_1$ regularization term and its penalization coefficient `alpha`.

```
1  def loss_fn_ctx(model, trajs, t_eval, ctx, alpha, ctx_, key):
2      """ Inner loss function """
3
4      trajs_hat, nb_steps = jax.vmap(model, in_axes=(None, None, None, 0))\
5                                  (trajs[:, 0, :], t_eval, ctx, ctx_)
6      new_trajs = jnp.broadcast_to(trajs, trajs_hat.shape)
7
8      term1 = jnp.mean((new_trajs-trajs_hat)**2)   # reconstruction
9
10     term2 = jnp.mean(jnp.abs(ctx))               # regularisation
11
12     loss_val = term1 + alpha*term2
13
14     return loss_val, (jnp.sum(nb_steps)/ctx_.shape[0], term1, term2)
```

LISTING 2: **Inner NCF loss function with vectorization support**

# E  TRAJECTORY VIZUALISATION

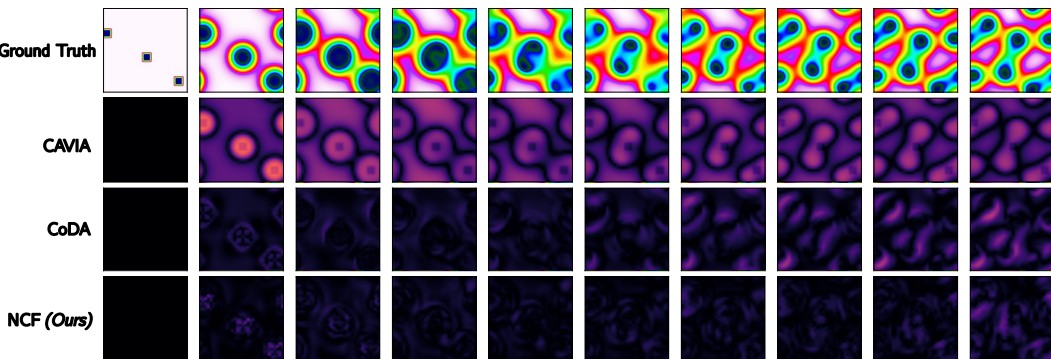

**FIGURE 6: Reconstruction absolute error of a trajectory from one environment during adaptation on the Gray-Scott system.**

