# OpenReview forum: "Neural Context Flows for Learning Generalizable Dynamical Systems"
_ICLR.cc/2024/Workshop/AI4DiffEqtnsInSci — AI4DiffEqtnsInSci @ ICLR 2024 Poster_

### Official Review · Reviewer_Gp84 · 2024-02-20
**Interesting and novel, although mainly empirical**

**Rating:** 7
**Confidence:** 3

**Review:**

This work presents Neural Context Flows (NCFs), a specialization of Neural ODEs to dynamical systems with unobserved (time-invariant) parameters. NCF introduces a (latent) context vector which modulates the behavior of the neural ODE. At training, a separate context vector is learned for each realization of the dynamical system along the parameters of the neural ODE. Importantly, during training, the neural ODE is linearized (by a first-order Taylor expansion around another context) with respect to the context, which seems to improve generalization with in-domain and out-of-domain parameters of the dynamical system.

### Strength
* The subject is interesting and well suited to the workshop.
* The goal of the paper is clearly presented and well introduced.
* To my knowledge, the approach is novel.
* The empiriacal results seem good, although I did not check them in details.

### Weaknesses
* As mentioned in the discussion, NCF lacks theoretical backing. At least an intuition as to why it works would profit the manuscript. My guess is that the linearization with respect to the context imposes a smooth modulation landscape.
* The quadratic cost of the training loss is also a concern, although it can likely be fixed with a Monte Carlo estimate (randomly selecting a fixed number of $j$ for each $e$ instead of all).
* Alternating between optimizing the weights $\theta$ and the contexts $\xi$ could lead to instabilities.

---

### Official Review · Reviewer_E7Cp · 2024-02-21
**Interesting topic, but lacks of appropriate details in mathematical background and analysis of results**

**Rating:** 4
**Confidence:** 5

**Review:**

Albeit proper references are provided about previous work on related topics, the reader should be provided with a self-contained description of the mathematical backgrounds, which the reviewer finds insufficient in the current version of the paper
For instance:
1. What is the functionality of the learnable context vectors? Why are they important?
2. What is the formula of the regularization function context? Are there multiple definitions available? Any guidelines on how to choose one over the other?

Additional discussions that the reviewer considers important, but are missing from the discussion about the context vectors:
1. How is the size of the context vectors chose? I see that you have a discussion about this in the appendix, but this is a serious limitation of the approach that should be mentioned in the main text as well.
2. Is it possible that trajectories provided by different environmemnts eventually collapse onto the same values of the context vector? What would this imply? No details are provided about this neither in the numerical results nor in the conclusions

---

### Meta-Review · Area_Chair_TvKK · 2024-02-21

**Recommendation:** Accept (Poster)

**Metareview:**

The reviewers mention that the subject of the paper on improving the generalization of NeuralODEs is well-suited for the workshop and its novelty. Both reviewers mention that mathematical detail and theoretical results are missing.  Given that it is a limited space workshop paper, it may be hard to include details that I would recommend to include in a full-length paper.  Overall, I would vote for acceptance.

---

### Decision · Program_Chairs · 2024-02-28

Accept (Poster)